# Proline to Threonine Mutation at Position 162 of NS5B of Classical Swine Fever Virus Vaccine C Strain Promoted Genome Replication and Infectious Virus Production by Facilitating Initiation of RNA Synthesis

**DOI:** 10.3390/v13081523

**Published:** 2021-08-02

**Authors:** Huining Pang, Ling Li, Hongru Liu, Zishu Pan

**Affiliations:** State Key Laboratory of Virology, College of Life Sciences, Wuhan University, Wuhan 430072, China; phn@whu.edu.cn (H.P.); concentrate1@aliyun.com (L.L.); 2019202040043@whu.edu.cn (H.L.)

**Keywords:** classical swine fever virus, 3′untranslated region, NS5B, RNA-dependent RNA polymerase, replication, initiation

## Abstract

The 3′untranslated region (3′UTR) and NS5B of classical swine fever virus (CSFV) play vital roles in viral genome replication. In this study, two chimeric viruses, vC/SM3′UTR and vC/b3′UTR, with 3′UTR substitution of CSFV Shimen strain or bovine viral diarrhea virus (BVDV) NADL strain, were constructed based on the infectious cDNA clone of CSFV vaccine C strain, respectively. After virus rescue, each recombinant chimeric virus was subjected to continuous passages in PK-15 cells. The representative passaged viruses were characterized and sequenced. Serial passages resulted in generation of mutations and the passaged viruses exhibited significantly increased genomic replication efficiency and infectious virus production compared to parent viruses. A proline to threonine mutation at position 162 of NS5B was identified in both passaged vC/SM3′UTR and vC/b3′UTR. We generated P162T mutants of two chimeras using the reverse genetics system, separately. The single P162T mutation in NS5B of vC/SM3′UTR or vC/b3′UTR played a key role in increased viral genome replication and infectious virus production. The P162T mutation increased vC/SM3′UTR_P162T_ replication in rabbits. From RNA-dependent RNA polymerase (RdRp) assays in vitro, the NS5B containing P162T mutation (NS5B_P162T_) exhibited enhanced RdRp activity for different RNA templates. We further identified that the enhanced RdRp activity originated from increased initiation efficiency of RNA synthesis. These findings revealed a novel function for the NS5B residue 162 in modulating *pestivirus* replication.

## 1. Introduction

Classical swine fever virus (CSFV) is causative agent of an economically important, highly contagious disease of pigs. CSFV is an enveloped, single-stranded positive-sense RNA virus. CSFV belongs to the genus *Pestivirus* in the family *Flaviviridae,* along with bovine viral diarrhea virus (BVDV) and border disease virus (BDV) [1]. The CSFV genome is approximately 12.3 kb containing one large open reading frame (ORF) flanked by two untranslated regions (5′UTR and 3′UTR) [2,3]. The ORF encodes a polyprotein of 3898 amino acids. The polyprotein is processed by cellular and viral proteases yielding 12 mature proteins: 4 structural proteins (Core, E^rns^, E1 and E2) and 8 non-structural proteins (N^pro^, p7, NS2, NS3, NS4A, NS4B, NS5A and NS5B) [4,5,6].

CSFV C strain is a lapinized live attenuated vaccine strain which was obtained by serial passages of a highly virulent CSFV strain in the rabbit in the 1950s [7,8]. CSFV C strain induces a typical fever response in the rabbit and a robust replication in the spleen of the rabbit [9]. The CSFV C strain is widely used as the live attenuated CSF vaccine and provides complete protection against virulent CSFV strains challenge [10,11,12].

The UTRs of CSFV are considered to be important elements for the translation and the replication of viral RNA [13,14], and also play an important role in the fever response in the rabbits induced by CSFV C strain [9]. The 3′UTR is proved to bind to NS5A and NS5B [15], and is probably involved in the initiation of the viral genome replication [16,17,18,19]. A 12-nt (CUUUUUUCUUUU) insertion existed in the 3′UTR of CSFV C strain, not in other wild-type CSFV strains [20,21]. The 12-nt insertion contributes greatly to the decrease of virus titer [22].

NS5B, an RNA-dependent RNA polymerase (RdRp), is essential for replication of the viral RNA genome [17,23,24]. NS5B from the family *Flaviviridae* carries out RNA synthesis through a de novo initiation mechanism [25,26]. The RNA synthesis catalyzed by NS5B is divided into representative steps: apo state, initiation and elongation [27]. The initiation of RNA synthesis contains a rapid formation of initial dinucleotide and a rate-limiting formation of trinucleotide [28]. Then RNA synthesis transitions to a processive elongation mode with the NS5B conformation opening up [29]. The crystal structure analysis revealed that the overall fold of CSFV NS5B contains the typical polymerase finger, palm, thumb domains and an additional N-terminal domain (NTD) [30,31]. A characteristic conserved Gly-Asp-Asp (GDD) motif of NS5B was observed among RdRps from positive-strand RNA viruses [32,33]. The finger domain is composed of two sections of the amino acid sequence (residues 136 to 313 and residues 351 to 408), which has been identified to determine the preference of RNA templates [31]. The fingertip region containing the N-terminal part of the finger domain (residues 136 to 160) and an insertion in the β-finger domain (residues 260 to 288) is implicated in NTP binding and interaction with RNA template in BVDV NS5B [34,35].

In the present study, the 3′UTR of CSFV C strain was substituted with 3′UTR of CSFV Shimen strain or BVDV NADL strain to generate vC/SM3′UTR and vC/b3′UTR, respectively. We investigated the characteristics of passaged viruses after serial passages. An adaptive Pro-to-Thr mutation (P162T) at position 162 of NS5B was identified in both the passaged vC/SM3′UTR and vC/b3′UTR. Site-mutated viruses were generated to investigate the influence of P162T mutation on viral replication and infectious virus production. The effect of P162T mutation on virus replication in the rabbits was also investigated. Wild-type NS5B protein (NS5B_WT_) and its mutant (NS5B_P162T_) were expressed and purified to characterize the catalytic activity in RdRp assays in vitro. Our results indicated that the residue 162 of CSFV NS5B, an amino acid close to the fingertip region, plays a vital role in viral genome replication and infectious virus production by regulating the initiation efficiency of the RNA synthesis.

## 2. Materials and Methods

### 2.1. Cells and Viruses

Porcine kidney-15 (PK-15) cells, swine kidney-6 (SK-6) cells, and BVDV NADL strain were obtained from the China Center for Type Culture Collection (CCTCC, Wuhan, Hubei, China). The PK-15 and SK-6 cells were maintained at 37 °C with 5% CO_2_ in Dulbecco’s modified Eagle’s medium (DMEM) (Invitrogen, Carlsbad, CA, USA) supplemented with 10% fetal bovine serum (FBS) (Natocor, Cordoba, Argentina). The CSFV C strain was generated from a cDNA clone (pSPT_I_/C) [36].

### 2.2. Construction of Infectious Chimeric cDNA Clones and Rescue of Viruses

Plasmid pSPT_I_/C, an infectious cDNA clone of CSFV C strain [36], was used to construct infectious chimeric cDNA clones containing CSFV Shimen strain or BVDV 3′UTR substitution. The 3′UTR fragment of CSFV Shimen strain or BVDV was amplified by overlap extension PCR using specific primers (C11152-F/C12072-R and SM-3′UTR-F/SM-3′UTR-R for Shimen 3′UTR, or C11152-F/C12072-R and NADL-3′UTR-F/NADL-3′UTR-R for BVDV 3′UTR). The fragments were cloned into pSPT_I_/C using the restriction enzymes *Sph* I and *Mlu* I to generate the infectious chimeric cDNA clones, pSPT_I_/C/SM3′UTR and pSPT_I_/C/b3′UTR, respectively. All constructs were sequenced to confirm their identities. The sequences of primers are listed in Table 1.

Virus rescue was performed as previously described [36]. Briefly, PK-15 cell monolayers in 6-well plates were transfected with 10 μL of Lipofectamine^®^2000 (Invitrogen) and 2.5 μg cDNA clone plasmid. The cell lysates were harvested by three cycles of freezing and thawing at 72 h post-transfection. After centrifugation at 5000× *g* for 10 min, the clarified supernatants were collected as the viral stocks and stored at −80 °C for subsequent assays. The viral stock was used to infect PK-15 cells to verify the recovery of infectious virus by immunofluorescence (IF) assay.

### 2.3. Continuous Passaging of Chimeric Viruses, Virus Titration, Plaque Assay and Growth Curve

The rescued chimeric virus aliquot (200 μL) was used to infect 30 mm diameter plates seeded with PK-15 cells. At 72 h post-infection, 200 μL culture fluid was transferred to a new plate containing PK-15 cells with 1.8 mL of culture medium. After every ten rounds of passages, the cell culture supernatant was collected and stored at −80 °C. After sequencing, the virus stocks from 11th, 21st, and 31st passages (p11, p21 and p31) were used for subsequent assays.

For virus titration, a serial 10-fold diluted viral stock in 100 μL was added to PK-15 cells in 96-well plates. The cells were fixed with 50% (*v*/*v*) methanol/acetone at 72 h post-infection and stained with an anti-NS3 antibody for IF assay. Viral titers were calculated using the method of Reed and Muench [37], and expressed as TCID_50_/mL.

Viral plaque assay was performed as previously described [38,39]. The SK-6 cell monolayers were infected at an MOI of 0.001, washed twice with PBS at 1 h post-infection, overlaid with 2% FBS medium containing 1.5% methylcellulose, and incubated at 37 °C for 96 h. The overlay was then removed, the cells were fixed with 50% (*v*/*v*) methanol/acetone and stained by immunohistochemistry assay with an anti-NS3 antibody using Two-Step IHC Detection Reagent (ZSGB-BIO, Beijing, China).

To determine the one-step growth curve of virus, PK-15 cells were infected with viruses at an MOI of 0.001 in a 12-well plate [40]. After adsorption for 2 h at 37 °C, the overlay was replaced with fresh medium containing 2% FBS and the cells were incubated at 37 °C. The viruses were harvested at indicated times and the viral titers were measured.

### 2.4. Immunofluorescence (IF) Assay

Immunofluorescence staining was performed using an anti-NS3 antibody as previously described [41]. The cells were fixed with 50% *v*/*v* methanol/acetone for 30 min at −20 °C and blocked with 3% bovine serum albumin (BSA) (Biosharp, Hefei, Anhui, China) for 30 min at 37 °C. Primary antibody (anti-NS3) incubation was conducted for 90 min at 37 °C with PBS containing 1% BSA, and secondary antibody (Alexa 488-conjugated goat anti-rabbit IgG, Jackson, West Grove, PA, USA) incubation was conducted for 60 min at 37 °C with PBS containing 1% BSA.

### 2.5. qRT-PCR

Viral RNA copy number was measured by quantitative reverse transcription PCR (qRT-PCR). Briefly, PK-15 cell monolayers were infected with the virus at an indicated multiplicity of infection (MOI). Total RNA was extracted from the cells at 6, 12, and 24 h post-infection using an RNApure Kit (Aidlab, Beijing, China), and 700 ng of total RNA was reverse transcribed using TRUEscript RT MasterMix (Aidlab). Then, 2 μL of cDNA was analyzed by qPCR using NovoStart^®^ SYBR qPCR SuperMix Plus (Novoprotein, Shanghai, China) with the specific primers qRT-PCR-NS5B-F/qRT-PCR-NS5B-R (Table 1). The data, presented as RNA copy number, were recorded from three independent experiments, and each experiment was performed in duplicate.

### 2.6. Construction of Infectious Chimeric cDNA Clones Harbouring Site Mutation

To construct the infectious clone of mutant pSPT_I_/C/SM3′UTR_P162T_ or pSPT_I_/C/b3′UTR_P162T_, the fragment containing C terminal of NS5A and NS5B region was amplified from pSPT_I_/C using specific primers C9522-F/C12047-R (Table 1). Then, the fragment was cloned to pClone007 simple vector (Tsingke, Tianjin, China). The NS5B P162T mutation was introduced using specific primers P162T-F/P162T-R (Table 1) to produce pClone007-NS5B_P162T_ [42]. The fragment containing the P162T mutation in pClone007-NS5B_P162T_ was used to replace the counterpart of pSPT_I_/C/SM3′UTR or pSPT_I_/C/b3′UTR using *Xba* I and *Spe* I. Both mutants were sequenced to confirm their identities. The mutated chimeric viruses were rescued and titrated.

### 2.7. Experimental Infection of Rabbits

The animal experiments were approved by the Institutional Animal Care and Use Committee of Wuhan University. Eight 14-week-old New Zealand White rabbits were divided into 3 groups and inoculated intravenously via the marginal ear vein with the vC/SM3′UTR, vC/SM3′UTR_P162T_ or DMEM, respectively. The rectal temperature of all rabbits was monitored every 6 h from 0 to 72 h post-infection to monitor the fever response. All rabbits were euthanized at 3 days post-inoculation [43]. The viral RNA copy numbers in the spleens of the rabbits were determined using qRT-PCR assay using the specific primers qRT-PCR-5′UTR-F/qRT-PCR-5′UTR-R and the probe CSFV-5′UTR-probe (Table 1) [44].

### 2.8. Expression and Purification of NS5B and Its Mutant

To prepare NS5B protein, the NS5B fragment with deletion of region encoding 24 C-terminal hydrophobic amino acids of CSFV C strain was amplified using primers C-NS5B-F/C-NS5B-R and T7-F/T7-R (Table 1), and then cloned into the expression plasmid pET28a. The P162T mutation was introduced using specific primers P162T-F/P162T-R (Table 1). Both constructs were confirmed by sequencing. Expression of NS5B and its mutant was performed as previously described [45,46]. Briefly, the constructs were transformed into *Escherichia coli* BL21-CodonPlus (DE3)-RIL strain, respectively, and cells were grown overnight at 37 °C in the LB medium with 50 μg/mL kanamycin (KAN50) and 25 μg/mL chloramphenicol (CHL25). The overnight culture was inoculated into 1 L of LB medium with KAN50 and CHL25. The cells were grown at 37 °C to an optical density of 0.6 at 600 nm and then were cooled to 20 °C, and isopropyl-β-D-thiogalactoside (IPTG) was added to a final concentration of 0.8 mM. After an additional 20 h incubation at 20 °C, the cells were harvested by centrifugation and stored at −80 °C for subsequent assay.

The cells were resuspended in a lysis buffer containing 300 mM NaCl, 50 mM Tris (pH8.0), 10 mM imidazole, 20% *v*/*v* glycerol, 1 U/mL DNase I, and 1× EDTA-free cocktail Protease Inhibitor (Yeason, Shanghai, China) and then were lysed by freezing and thawing followed by sonication. The lysate was clarified by centrifugation for 30 min at 15,000× *g* in a JLA-16.250 rotor (Beckman, Brea, CA, USA). The clarified lysate was loaded onto a HispurTM Ni-NTA Resin column (Thermo Scientific, Waltham, MA, USA). After washing with a wash buffer (300 mM NaCl, 50 mM Tris (pH 8.0), 75 mM imidazole, 20% *v*/*v* glycerol), the protein was eluted with elution buffer (300 mM NaCl, 50 mM Tris (pH 8.0), 250 mM imidazole, 20% *v*/*v* glycerol). Purified NS5B and its mutants were examined by SDS-PAGE. The fractions containing purified protein were pooled and dialyzed in 500 mM NaCl, 50 mM Tris (pH7.5), 20% *v*/*v* glycerol, and 5 mM dithiothreitol (DTT). The pooled fractions were then concentrated to approximately 120 μM, flash frozen with liquid nitrogen and stored at −80 °C in small aliquots.

### 2.9. RNA Template Preparation

The chemically synthesized 30-nt template strand (T30) purification, self-annealing, and annealing with a GG dinucleotide primer bearing a 5′-phosphate (P2) at a 1:1.25 molar ratio to yield the T30/P2 construct were performed as described previously [30]. The homopolymeric poly(C) RNA was purchased from Sigma (St. Louis, MO, USA). The 228-nt positive strand RNA of CSFV Shimen strain 3′UTR ((+) CSFV/3′UTR) was transcribed in vitro using an amplified fragment from pSPT_I_/C/SM3′UTR with specific primers T7-3′UTR-F/3′UTR-R (Table 1) and the transcripts were purified by 6% (*w*/*v*) polyacrylamide/7 M urea gel electrophoresis. The RNA was extracted using an RNApure Kit (Aidlab). The RNA concentration was measured using UV spectrophotometry and stored at −80 °C.

### 2.10. RNA-Dependent RNA Polymerase (RdRp) Assay

The de novo RdRp assay using T30 template was performed using the dinucleotide (P2)-driven reactions [30,47]. A 20 μL reaction mixture containing 4 μM T30/P2, 15 μM extra P2, 6 μM NS5B, 300 μM ATP, 300 μM UTP, 30 mM NaCl, 50 mM Tris (pH7.0), 5 mM MgCl_2_, and 5 mM DTT was incubated at 30 °C for indicated times. For RdRp assays, reaction quenching, sample processing, denaturing polyacrylamide gel electrophoresis (PAGE) and RNA visualization by Stains-All (Sigma-Aldrich, St. Louis, MO, USA) staining were performed as previously described using the T30 template [48]. To quantify the intensity of 9-nt RNA product (P9), images of Stains-All based gels were taken with ChemiDoc MP (Bio-Rad, Hercules, CA, USA) and analyzed by Image Lab 5.2.1.

To evaluate the elongation activity of RdRp, a single-nucleotide elongation assay was performed as previously described [48]. Briefly, the reaction was performed in a two-step format. The reaction in the first step for 9-nt RNA synthesis proceeded for 50 min, and the mixture was then centrifuged at 13,000× *g* for 1 min. The pellet was harvested and washed twice by the reaction buffer (30 mM NaCl, 50 mM Tris (pH 7.0), 5 mM MgCl_2_, 5 mM DTT) and then resuspended in 18 μL modified reaction buffer with 200 mM NaCl. In the second step, 2 μL CTP was supplemented to the resuspended mixture for a final concentration 300 μM or 5 μM to allow the single-nucleotide elongation at 30 °C, and the reaction was quenched immediately (“0 min”) or after 1 min. To accurately measure the reaction rate, an extra reaction was conducted at 0 °C under 5 μM CTP and the reaction was quenched immediately (“0 s”) or at 20 s, 40 s, 60 s.

To evaluate the stability of elongation complex (EC), a high-salt (500 mM NaCl) incubation challenge was added into the single nucleotide elongation assay. The resuspended reaction mixture was incubated in the high-salt reaction buffer at 37 °C for indicated times (0–72 h), and then CTP was supplied at 300 μM final concentration to trigger the P9-to-P10 conversion by P9-containing EC (EC9) that survived the incubation. The reaction proceeded for 1 min at 30 °C. After subtracting the intensity of the P10 misincorporation product (P10_m_) from the first step, the intensity fraction of P10 ([P10_int_ − P10_m, int_]/[P9_int_ + P10_int_ − P10_m, int_]) was used to calculate the fraction of the EC9 that survived the high-salt incubation. The fraction of P10 intensity value was fitted to one phase decay model to estimate the inactivation rate.

To test RdRp initiation efficiency, a P2 to the 3-nt product (P3) conversion assay was performed as previously described [47]. Briefly, ATP was supplied as the only NTP substrate and the final concentrations of ATP were 100, 200, 400, 800, and 1200 μM. A high P2:T30 ratio (20:1) was used for reaction to achieve multiple turnovers within a reasonable duration. The reactions were conducted at 30 °C for different times. Samples from the same mixture were loaded on different gels to normalize the intensities of product bands and the normalized intensity of P3 was then used to calculate the relative reaction rates that were fitted to the Michaelis–Menten equation.

Next, we further determined the RdRp activity for other RNA templates as previously described [49,50]. The accumulation of pyrophosphate (PPi) was measured during the RNA polymerization reaction using a phosphate-based colorimetric method with malachite green-molybdate as the color-developing reagent. The reactions were carried out in 25 μL solution containing 2 μM NS5B, 20 mM Tris (pH7.5), 2.5 mM MgCl_2_, 5 mM DTT, 40 U/mL RNase Inhibitor, 20 U/mL thermostable pyrophosphatase (PPase), 20% glycerol, 800 μM GTP, and 500 ng poly(C) at 37 °C. For the template of 0.1 μM (+) CSFV/3′UTR, 200 μM NTPs were supplied as the substrate. The reactions were quenched by heating at 70 °C for 3 min, where PPi was converted to two molecules of inorganic phosphate (Pi) by PPase simultaneously. The concentration of Pi was detected by mixing 10 μL of the reaction mixture with 30 μL Milli-Q water and 100 μL of the malachite green-molybdate reagent. The phosphomolybdate complexes were measured by the absorbance at 630 nm by Multiskan MK3 (Thermo Scientific). The concentration of Pi was quantified using a phosphate standard curve obtained from 0 to 100 μM NaH_2_PO_4_ solution.

### 2.11. Statistical Analysis

Statistical analyses of the data were performed using Student’s *t*-test or one-way analysis of variance (ANOVA). A *p*-value less than 0.05 was considered significant.

## 3. Results

### 3.1. Rescue of Chimeric CSFV Vaccine C Strains Containing Pestivirus 3′UTR Substitution and Characteristics of Passaged Viruses

To investigate the influence of *pestivirus* 3′UTR on viral characteristics and genome genetic stability, we substituted the 3′UTR of CSFV vaccine C strain with 3′UTR from CSFV Shimen strain or BVDV NADL strain to generate two chimeric infectious cDNA clones, pSPT_I_/C/SM3′UTR and pSPT_I_/C/b3′UTR, and the chimeric viruses vC/SM3′UTR and vC/b3′UTR were rescued from chimeric infectious cDNA clones, respectively (Figure 1a). The culture supernatants collected at 72 h post-transfection were used to infect PK-15 cells. After incubation of 24 h, the infected PK-15 cells were assessed by IF assay using an anti-NS3 antibody for detection of viral protein expression (Figure 1b). Data showed that the infectious viruses, vC/SM3′UTR and vC/b3′UTR, were successfully obtained.

Next, the rescued vC/SM3′UTR and vC/b3′UTR were passaged in PK-15 cells every 3 days and both chimeric viruses were passaged for a total of 31 passages. The viruses amplified in PK-15 cells (p3, 3rd passaged) were used as the parent viruses for comparison. Viral RNA copy numbers and virus titers were measured every 10 passages in PK-15 cells. For vC/SM3′UTR, significantly increased RNA copy numbers were observed between p3 and p11, p11 and p21 at 24 h post-infection, and increased virus titers from p3 to p11 and p11 to p21 were observed (Figure 1c,d). However, for vC/b3′UTR, both significantly increased virus replication and infectious virus production were observed from p11 to p21 and p21 to p31, and significantly increased RNA copy numbers between p3 and p11 appeared at 24 h post-infection (Figure 1c,d). Moreover, the one-step growth curve of each virus exhibited a similar proliferation trend (Figure 1e). The serial passages also resulted in increased plaque sizes for both vC/SM3′UTR and vC/b3′UTR (Figure 1f).

To evaluate virus genetic stability, the whole genomes of 31st passaged vC/SM3′UTR and vC/b3′UTR were sequenced. The mutation sites observed in 31st passaged viruses were further sequenced in 11th and 21st passaged viruses for comparison (Table 2). Compared to parent viruses, the 5′UTR and 3′UTR sequences of 31st passaged vC/SM3′UTR and vC/b3′UTR had no change. Five amino acid mutations were detected in the polyprotein of 31st passaged vC/SM3′UTR or vC/b3′UTR, respectively. The mutations S476R, M979K and P3342T occurred in both 31st passaged vC/SM3′UTR and vC/b3′UTR. The mutations L2671M and G3431S were only observed in 31st passaged vC/SM3′UTR. T745I and N2494H occurred only in 31st passaged vC/b3′UTR. For vC/SM3′UTR, M979K mutation in 11th passaged virus resulted in the significantly increased viral RNA copy number and virus titer compared to 3rd parent virus. L2671M, P3342T, and M979K mutations in 21st passaged virus may have synergistically contributed to the change of virus characteristics. S476R and G3431S mutations in the 31st passaged virus failed to affect virus characteristics compared to the 21st passaged virus. For vC/b3′UTR, P3342T and the slight M979K mutations in the 21st passaged virus resulted in the significantly increased viral RNA copy number and virus titer compared to the 11th passaged virus. S476R, T745I, M979K, and N2494H mutations in the 31st passaged virus may have synergistically contributed to the change of virus characteristics compared to the 21st passaged virus. We previously demonstrated that the mutation M979K in E2 increased infectious virus production, cell to cell spread and virulence for pig [38]. We reasonably speculated that the mutation P162T (P3342T in polyprotein) in NS5B plays a vital role in regulating virus replication.

### 3.2. Proline to Threonine Mutation at Position 162 of NS5B Increased Chimeric CSFVs RNA Replication and Infectious Virus Production

To investigate the effect of NS5B P162T mutation on virus characteristics, the single-site mutant was constructed based on the infectious cDNA clone pSPT_I_/C/SM3′UTR or pSPT_I_/C/b3′UTR, respectively (Figure 2a). The mutant viruses (vC/SM3′UTR_P162T_ and vC/b3′UTR_P162T_) were rescued. Viral RNA copy numbers and virus titers were determined in PK-15 cells at an MOI of 0.01. Data showed that the mutants vC/SM3′UTR_P162T_ and vC/b3′UTR_P162T_ displayed significantly increased viral RNA copy numbers and virus titers compared to parent vC/SM3′UTR and vC/b3′UTR, respectively (Figure 2b,c). 

### 3.3. The Influence of P162T Mutation on Fever Response and Virus Replication in Rabbits

Rabbit is a useful animal model for evaluating replication efficiency of CSFV C strain in vivo [9,43,51]. To investigate the effect of NS5B P162T mutation on virus in vivo, three groups of eight rabbits were inoculated with vC/SM3′UTR, vC/SM3′UTR_P162T_ or Dulbecco’s modified Eagle’s medium (DMEM), respectively. The group inoculated with DMEM was used as a negative control. Rectal temperature was recorded every 6 h after inoculation until 72 h post-inoculation. In contrast to DMEM, vC/SM3′UTR, vC/SM3′UTR_P162T_ induced different degrees of fever responses in rabbits. The maximum average temperature of vC/SM3′UTR_P162T_ was 0.7 °C lower than that of vC/SM3′UTR (Table 3). All rabbits were euthanized for determining viral RNA copy numbers in the spleens by qRT-PCR. The significantly increased viral RNA copies in the spleens of rabbits infected with vC/SM3′UTR_P162T_ were observed compared to that with vC/SM3′UTR (Table 3). 

### 3.4. NS5B P162T Mutation Increased RNA-Dependent RNA Polymerase Activity In Vitro

NS5B, an RNA-dependent RNA polymerase, is responsible for *pestivirus* genome replication. To investigate the influence of NS5B P162T mutation on RdRp activity, we expressed an NS5B construct comprising residues 1–694 with deletion of the C-terminal 24aa hydrophobic region as the wild-type (NS5B_WT_). P162T mutation was introduced into NS5B_WT_ to generate a mutant NS5B_P162T_. Both NS5B_WT_ and NS5B_P162T_ were expressed and purified. After being judged by SDS-PAGE (Appendix A), purified proteins were used for subsequent RdRp assays in vitro.

To measure de novo RdRp activity, a GG dinucleotide (P2) driven RdRp assay was performed as previously described [30,48,52]. We compared the RdRp activity of NS5B_WT_ and NS5B_P162T_ using the assay with a 30-nt RNA template (T30). T30 and P2 were annealed to form T30/P2 RNA substrate, and a 9-nt product (P9) was generated when ATP and UTP were provided as the only two NTP substrates (Figure 3a). The P9 product was detected at different time points by denaturing urea polyacrylamide gel electrophoresis (PAGE) (Figure 3b) and the apparent P9 conversion rate constant (*k_conv_*) was calculated by the intensity of P9 as a function of time (Figure 3c). Data showed that the *k_conv_* value of NS5B_P162T_ was more than twice that of NS5B_WT_ (0.03072 min^−1^ vs. 0.01433 min^−1^), indicating that the P162T mutation in NS5B upregulated de novo RdRp activity.

We next measured the RdRp activity using different RNA templates with a simple colorimetric assay [49,50]. To detect the polymerization reaction rate, the reaction product pyrophosphate (PPi) was measured using a phosphate-based colorimetric method by hydrolyzing PPi to two inorganic phosphates (Pi). The accumulation of Pi was detected at different time points to yield an apparent synthetic rate constant (*k_synt_*) of RNA synthesis. For the poly(C) template, the values of *k_synt_* were 0.02637 min^−1^ and 0.05529 min^−1^ characterized by NS5B_WT_ and NS5B_P162T_, respectively (Figure 3d). For viral template, (+) CSFV/3′UTR, the values of *k_synt_* of NS5B_WT_ and NS5B_P162T_ were 0.02189 min^−1^ and 0.02980 min^−1^, respectively (Figure 3e). These data indicated that the P162T mutation increased NS5B RdRp activity for different RNA templates in vitro.

### 3.5. The RdRp Elongation Activity and Stability of Elongation Complex (EC) Were Not Affected by NS5B P162T Mutation

RNA synthesis catalyzed by NS5B RdRp can be divided into three phases: apo state, initiation, and elongation [27]. To accurately understand the catalytic mechanism of NS5B_P162T_, we first tested the RdRp elongation activity and stability of NS5B_P162T_ elongation complex (EC). A P9 to P10 single nucleotide elongation assay was used to analyze the elongation activity of EC. The EC containing the 9-nt product (EC9) in de novo RdRp assay was employed to assess elongation activity by using a high-salt or heparin challenge strategy, which has been widely used in nucleic acids polymerase studies [30,52,53]. The single nucleotide elongation assay was designed in a two-step format. ATP and UTP were supplied to generate the EC9 in the first step. The proportions of P10 misincorporation product (P10_m_) in all assays were similar (Figure 4b–d, lanes with S1). In the second step, high-salt (200 mM NaCl) was introduced to block initiation events after removal of the originally supplied ATP and UTP, and then CTP was added as the only substrate to probe EC9-drived formation of EC10 (Figure 4a). We compared the P9 to P10 conversion under high and low CTP concentrations at 30 °C for NS5B_WT_ and NS5B_P162T_. At 300 μM CTP concentration, the conversion rates of P9 to P10 catalyzed by NS5B_WT_ and NS5B_P162T_ were 80.3% and 80.4% at “0 min”, and 89.1% and 94.1% at 1 min, respectively (Figure 4b). At 5 μM CTP concentration, decreased conversion rates of P9 to P10 were observed and the conversion rates catalyzed by NS5B_WT_ and NS5B_P162T_ were 32.1%, 32.6% at “0 min” and 92.4%, 93.8% at 1 min, respectively (Figure 4c). We determined the conversion rates catalyzed by NS5B_WT_ and NS5B_P162T_ under 5 μM CTP concentration at 0 °C. The conversion rates of NS5B_WT_ and NS5B_P162T_ at detailed time points (“0s”, “20s”, “40s”, “60s”) were almost identical (Figure 4d). Data suggested that the elongation activity of NS5B EC was not affected by the P162T mutation.

To further investigate the stability of EC, a high-salt (500 mM NaCl) incubation challenge was added into the single nucleotide elongation assay (Figure 5a). The proportion of EC9 which survived the high-salt challenge reflected the stability of EC. Conversion rate of P9 to P10 was the proportion of EC9 survival because only the surviving EC9 had the capacity to generate EC10. By measuring the conversion rates from P9 to P10 catalyzed by NS5B_WT_ and NS5B_P162T_, we found that the EC9 comprised of NS5B_WT_ or NS5B_P162T_ exhibited similar inactivation rate constants (0.04364 h^−1^ and 0.04185 h^−1^) and comparable half-life period (15.88 h and 16.56 h) (Figure 5c). Therefore, the P162T mutation had no effect on the stability of the RdRp elongation complex.

### 3.6. The Increased RdRp Activity of NS5B_P162T_ Mutant Resulted from Enhanced Initiation Efficiency of RNA Synthesis

To investigate the RdRp activity during initiation stage, the relative catalytic rates of NS5B_WT_ or NS5B_P162T_ were measured using a P2 to P3 conversion assay under five different ATP concentrations (Figure 6a,b). Experimental data were used to fit a Michaelis–Menten equation (Figure 6c). The P3 band intensities were controlled to be within the linear range of Stains-All based staining method by optimizing the reaction time [48]. The Michaelis constant (*K_M_*) values of NS5B_WT_ and NS5B_P162T_ were almost consistent (681.8 μM vs. 653.6 μM), while the relative maximum conversion rate constant (*V_max_*) value of NS5B_P162T_ was more than twice the value of NS5B_WT_ (0.3507 min^−1^ vs. 0.1534 min^−1^). The relative specificity constants (*k_cat_/K_M_*) of NS5B_WT_ and NS5B_P162T_ were 3.75 × 10^−5^ and 8.94 × 10^−5^ (Figure 6c). Data indicated that NS5B P162T mutation increased the RdRp initiation efficiency.

## 4. Discussion

CSFV and BVDV belong to the genus *Pestivirus*. Clinically, pigs can be simultaneously infected with CSFV, BVDV-1, and BVDV-2 [54]. Some genome elements and viral proteins of CSFV can be substituted with their counterparts in genus *Pestivirus* [55,56,57]. The 3′UTRs from CSFV C strain and BVDV NADL strain have similar predicted secondary structures [58,59], and the 3′UTR substitution has not effect on replication across species in chimeric C strains [9]. In this study, the chimeric CSFV C strain containing BVDV NADL 3′UTR substitution was successfully rescued. Therefore, the 3′UTR substitution of CSFV with BVDV NADL 3′UTR is feasible.

In our study, five amino acid mutations were identified in 31st passages of both passaged vC/SM3′UTR and vC/b3′UTR. For vC/SM3′UTR, S476R and G3431S mutations in 31st passaged virus had no influence on the viral genome replication and infectious virus production compared to 21st passaged virus. However, the single Ser-to-Arg mutation at position 476 in the CSFV E^rns^ resulted in the virus binding to cells from a heparan sulfate (HS)-independent mode to an HS-dependent mode [60,61]. For vC/b3′UTR, S476R, T745I, and N2494H in 31st passaged vC/b3′UTR resulted in significantly increased viral genome replication and infectious virus production, and the T745I mutation probably played a vital role. T745I has been identified to enhance the infectious virus production, cell to cell spread, and pathogenicity [38]. Compared with other mutations, the M979K and P3342T occurred earlier in both passaged vC/SM3′UTR and vC/b3′UTR. Effect of P3342T mutation in CSFV BS5B is further addressed.

Alignment of amino acid sequences revealed that proline at position 3342 of CSFV polyprotein only existed in lapinized attenuated strains including vaccine C, Riems, LPC, and HCLV-India strains. In contrast, other CSFV strains, such as highly virulent and moderately virulent strains and another attenuated strains, the residue 3342 of polyprotein is threonine (data not shown). A previous study showed that the proline at position 3342 of CSFV C strain remained unchanged even in the 80th generation viruses [40]. In our study, the P3342T mutation was associated with the 3′UTR substitution and vC/SM3′UTR caused more severe fever response than vC/SM3′UTR_P162T_ in rabbits, suggesting that the proline at position 3342 seems to originate from adaptive evolution of CSFV in rabbits. This discovery provided new insight to reveal the mechanism of CSFV proliferation across interspecific barriers in non-susceptible animals. On the other hand, the P3342T mutation increased the infectious virus production in vitro and even in vivo, and had potential practical value in upstream CSVF viral vaccine manufacturability.

In our study, the vC/SM3′UTR induced a severe rabbit fever response with a maximum average temperature of 41 °C (Table 3). Previous reports demonstrated that the chimeric CSFV C strain harboring Shimen 3′UTR substitution failed to induce the fever response in the rabbits [9], and that the amino acid mutations in viruses or the substitutions of structure proteins could influence the fever response induced by C strain in the rabbits [40,62,63,64]. Nine different amino acids in structure and non-structure proteins existed between our CSFV C strain and the C-strain/HVRI (GenBank: AY805221.1) by aligning the amino acid sequences of polyproteins (Appendix A), suggesting that the fever response in the rabbits induced by CSFV C strains was probably regulated by both viral genome structure and amino acid composition.

The mutant NS5B_P162T_ exhibited increased RdRp activity compared to NS5B_WT_ using three types of RNA templates: T30, poly(C), and (+) CSFV/3′UTR [18,30,65]. We further demonstrated that the similar elongation activity and stability of EC were observed from NS5B_WT_ and NS5B_P162T_. However, mutant NS5B_P162T_ displayed enhanced initiation efficiency of RNA synthesis independent of initial NTP binding because of the similar Michaelis constant *K_M_* [30]. The residue 162 is located at the N-terminal of an α-helix in the index finger domain of CSFV NS5B structure close to the fingertip region [31,34]. The fingertip connects the finger domain with the thumb domain and results in the molecule having a unique globular shape with the active site cavity completely encircled [30,31,66]. The flexibility of the fingertip region seems to be an intrinsic property of the polymerase, which is important for the polymerase activity [34,67,68]. The fingertip region has been proposed to be involved in RNA template and NTP binding [31,34,69,70]. Mutation of residue 141 in the fingertip region of HCV NS5B corresponding to residue 163 in BVDV NS5B and CSFV NS5B resulted in drastically reduced polymerase activity [31,34,71]. HCV and BVDV replications were inhibited by the inhibitors which prevent the fingertips from moving [34,68]. The effect of P162T mutation in CSFV NS5B on initiation efficiency of RNA synthesis could be involved in the conformation change of the fingertip region. Previous studies identified that serial passages of chimera CSFV/CE23′UTR containing C strain 3′UTR substitution resulted in alanine to threonine mutation of the residue 392 located in the tip of the middle finger near the residue 162 [31,72]. The relative positions of residue 162 and residue 392 are fixed both in CSFV and BVDV NS5B structures (Appendix A) [30,31,34,35]. The functional regulation between residue 162 in the index finger and residue 392 in the middle fingertip of NS5B may be meaningful and deserves attention.

In summary, we identified the mutation P162T in CSFV NS5B and addressed the effects of this point mutation on NS5B RdRp activity and virus replication. The P162T mutation occurred in the continuous passages of two chimeric viruses with 3′UTR substitution and influenced viral genomic RNA synthesis by changing initiation efficiency in vitro or in vivo. These findings help us understand the impact of 3′UTR on virus adaptive evolution and brings insights into the intramolecular regulation mechanism of index finger domain of NS5B structure on RdRp activity.

## Figures and Tables

**Figure 1 viruses-13-01523-f001:**
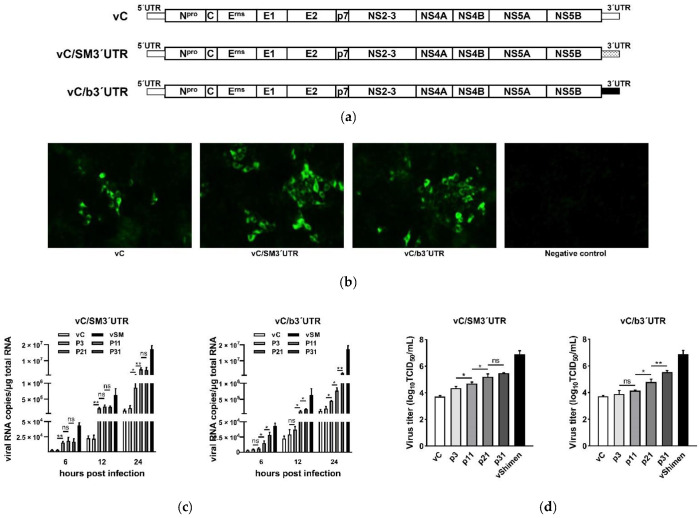
Rescue of chimeric classical swine fever virus (CSFV) vaccine C strains containing *pestivirus* 3′UTR substitution and characteristics of passaged viruses. (**a**) Schematic representation of the cDNA clones of chimeric CSFV C strains containing 3′UTR substitution. (**b**) Rescue of infectious chimeric viruses vC/SM3′UTR and vC/b3′UTR. PK-15 cells infected with the culture supernatant of transfected cells were analyzed by immunofluorescence (IF) staining with an NS3-specific antibody at 24 h post-infection. The microscopy images were captured at 100× magnification. (**c**) Genome replication efficiency of passaged viruses. PK-15 cells were infected with passaged chimeric viruses p3, p11, p21, and p31 at an MOI of 0.001, respectively. The total RNA was extracted from the infected cells at the indicated times and viral RNA copy numbers were determined by qRT-PCR. The results were recorded from three independent experiments. ns, no significant; *, *p* < 0.05; **, *p* < 0.01. (**d**) Virus titers of passaged viruses. PK-15 cells were infected with passaged viruses at an MOI of 0.01. Virus titers were measured at 72 h post-infection by titration of the TCID_50_/mL. Data are presented as the mean values ± SD from three independent experiments. The comparisons of the values were performed using Student’s *t*-test. ns, no significant; *, *p* < 0.05; **, *p* < 0.01. (**e**) Growth kinetics of passaged viruses. PK-15 cells were infected with passaged viruses at an MOI of 0.001. Data are presented as the mean values ± SD from three independent experiments. (**f**) Plaque morphology of passaged viruses. SK-6 cells infected with passaged viruses were analyzed by immunohistochemical staining with an NS3-specific antibody at 96 h post-infection. The images of plaque morphology were obtained at 1× magnification.

**Figure 2 viruses-13-01523-f002:**
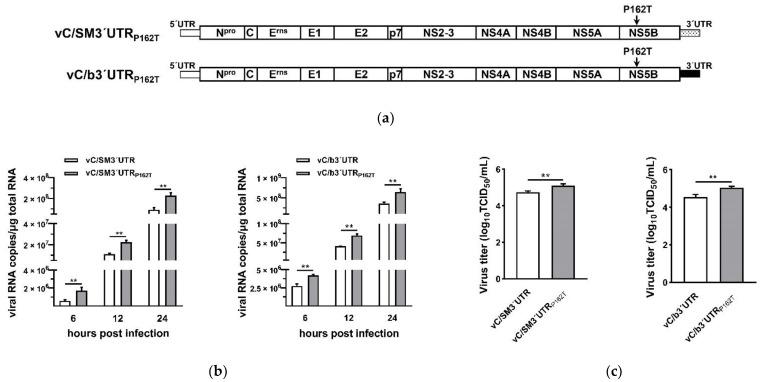
Viral genome replication and virus titers of chimeric CSFV C strains containing NS5B P162T mutation. (**a**) Schematic representation of the cDNA clones of vC/SM3′UTR_P162T_ and vC/b3′UTR_P162T_. (**b**) Genome replication efficiency of vC/SM3′UTR_P162T_ and vC/b3′UTR_P162T_ mutants. PK-15 cells were infected with chimeric viruses and their P162T mutants at an MOI of 0.01. The total RNA was extracted from the infected cells at the indicated time and viral RNA copy numbers were determined by qRT-PCR. Data were recorded from three independent experiments. (**c**) Virus titers of vC/SM3′UTR_P162T_ and vC/b3′UTR_P162T_ mutants. PK-15 cells were infected with the viruses at an MOI of 0.01 and the culture supernatants collected at 72 h post-infection were used for virus titration. Virus titers were expressed as the TCID_50_/mL. Data are represented as the mean values ± SD from three independent experiments. **, *p* < 0.01.

**Figure 3 viruses-13-01523-f003:**
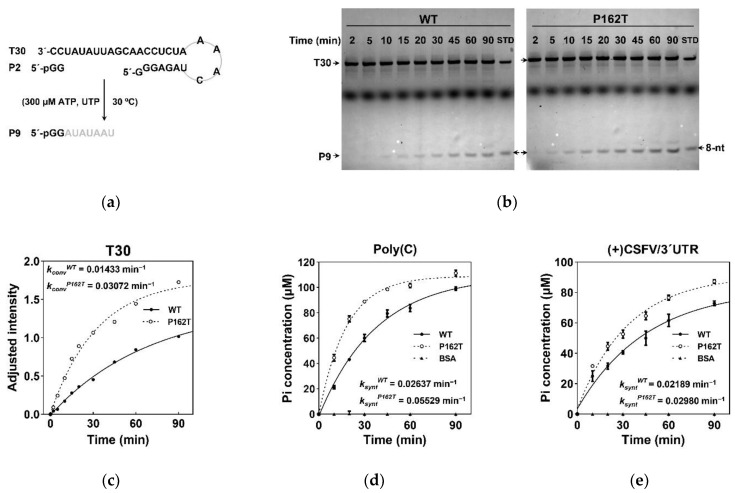
The NS5B RdRp activity in vitro. (**a**) Schematic diagram of the de novo RdRp assay in vitro. T30/P2 was used as the RNA substrate. The template strand T30 directed a 7-nt (grey) extension of the dinucleotide primer P2 (black) to produce a 9-nt product (P9) when ATP and UTP were supplied as the only NTP substrates. (**b**) Denaturing urea-PAGE analysis of the de novo RdRp assay. The P9 accumulation was monitored at the indicated time points for NS5B_WT_ and NS5B_P162T_ constructs. The oval-shaped band between T30 and P9 is the Xylene Cyanol FF. An 8-nt marker was used as the standard sample for quantitation (set to 1). The 8-nt marker bearing hydroxyl groups at the 5′-end was chemically synthesized and therefore migrated slower than the P9 bearing a 5′-phosphate. STD, standard sample containing T30 and 8-nt marker. (**c**) Comparison of de novo RdRp activity for NS5B_WT_ and NS5B_P162T_. The relative intensity of P9 product was plotted as a function of time to estimate the apparent P9 conversion rate constant (*k_conv_*) by fitting to a one phase exponential association model. (**d**,**e**) Comparison of RdRp activity for NS5B_WT_ and NS5B_P162T_ by colorimetric assay using the poly(C) and (+) CSFV/3′UTR templates. The Pi concentration was plotted as a function of time to fit to a one phase exponential association model and the apparent synthetic rate constant (*k_synt_*) was estimated.

**Figure 4 viruses-13-01523-f004:**
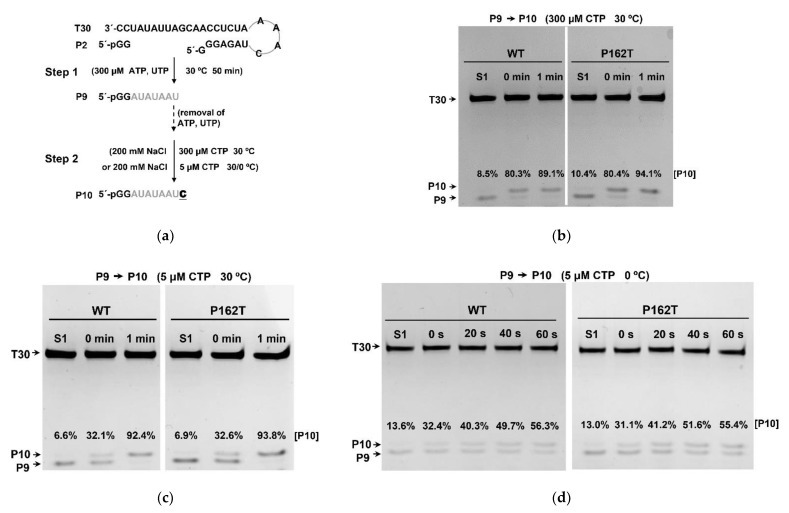
The elongation activity of RdRp elongation complex. (**a**) Schematic diagram of the P9 to P10 single nucleotide elongation assay. The first step of the assay was identical to the de novo RdRp assay. In the second step, CTP was supplied as the only NTP substrate and the NaCl concentration was raised to 200 mM after the removal of ATP and UTP. (**b**,**c**) Comparisons of elongation activity for NS5B_WT_ and NS5B_P162T_ at 300 or 5 μM CTP at 30 °C. (**d**) Comparison of elongation activity for NS5B_WT_ and NS5B_P162T_ at 5 μM CTP at 0 °C. The fraction of P10 intensity is shown in each lane. S1, sample from step 1.

**Figure 5 viruses-13-01523-f005:**
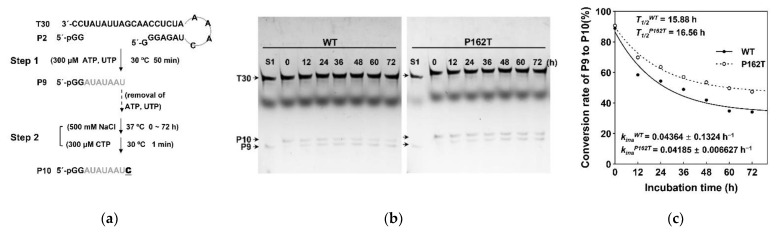
The stability of the RdRp elongation complex. (**a**) Schematic diagram of the stability assay. The first step of the assay was identical to the de novo RdRp assay. In the second step, a high-salt (500 mM NaCl) challenge was added before the supply of CTP. (**b**) Denaturing urea-PAGE analysis of the P9 to P10 conversion. The oval-shaped band between T30 and P10 is the Xylene Cyanol FF. S1, sample from step 1. (**c**) Comparison of stability of EC for NS5B_WT_ and NS5B_P162T_. The conversion rate of P9 to P10 was plotted as a function of incubation time to estimate the apparent EC inactivation rate constant (*k_ina_*) and half-life period (*T_1/2_*) by fitting to a one phase exponential decay model.

**Figure 6 viruses-13-01523-f006:**
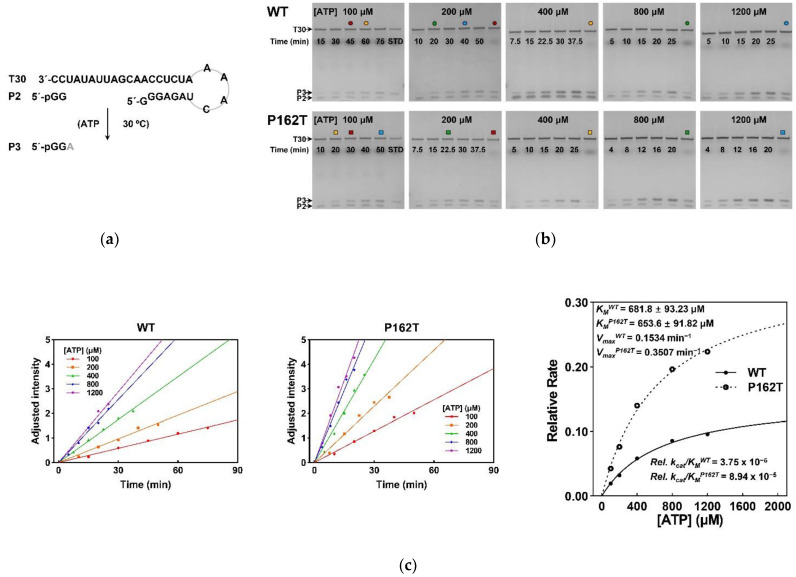
Characteristics of the initiation kinetics on NS5B_WT_ and NS5B_P162T_. (**a**) Schematic diagram of the P2 to P3 conversion assay. (**b**) Denaturing urea-PAGE analysis of the P3 formation under different ATP concentrations by NS5B_WT_ and NS5B_P162T_ constructs. The lanes marked with same color-coded icons represented the banks from the same sample which was used to correlate band intensity in different gels. STD, standard sample containing P3 from the pre-experiment. (**c**) Comparison of the initiation efficiency for NS5B_WT_ and NS5B_P162T_. The relative reaction rates of NS5B_WT_ and NS5B_P162T_ constructs were estimated by the equation about the adjusted intensity of the P3 as a function of time at five ATP concentrations. The rate values were then fit to the Michaelis–Menten equation to generate the Michaelis constant (*K_M_*), relative maximum reaction rate (*V_max_*), and relative specificity constant (*k_cat_/K_M_*).

**Table 1 viruses-13-01523-t001:** Primers used in this study.

Primer	Sequence (5′–3′)
C11152-F	GCAACAGCATGCTAAATGTC
C12072-R	CTCATGCCCCTCTCCCTATCAGC
SM-3′UTR-F	AGAGGGGCATGAGCGCGGGTAACCCG
SM-3′UTR-R	CGACGCGTCGGAGTACTGGTCGACCTCCGAAGTTGGGGGGGAGGGCCGTTAGGAAATTAC
NADL-3′UTR-F	GAGAGGGGCATGAGACAAAATGTATATATTG
NADL-3′UTR-R	CGACGCGTCGGAGTACTGGTCGACCTCCGAAGTTGGGGGGGAGGGGGCTGTTAGAGGTC
C9522-F	GACCCACTAGAAGTGAGAGATATG
C12047-R	CATCATCATGACTCTCAGCC
P162T-F	GGGCCCAGACAGACACAACCAACTTCCACCAAGCAA
P162T-R	GTGGAAGTTGGTTGTGTCTGTCTGGGCCCTAACCA
C-NS5B-F	AGGAGATATACCATGAGTAATTGGGTGATGCAAGAAG
C-NS5B-R	CCGCTCGAGATTGTACCTGTCTGTCCCTTG
T7-F	TAGGAAGCAGCCCAGTAGTAG
T7-R	CATCACCCAATTACTCATGGTATATCTCCTTCTTAAAGTT
qRT-PCR-NS5B-F	ATCTGCCTACAAGGAAGTCATCGG
qRT-PCR-NS5B-R	CCAGTTGCCCTCTTTAACACCCATA
qRT-PCR-5′UTR-F	ATGCCCATAGTAGGACTAGCA
qRT-PCR-5′UTR-R	CTACTGACGACTGTCCTGTAC
CSFV-5′UTR-probe	6FAM-TGGCGAGCTCCCTGGGTGGTCTAAGT-BHQ1
T7-3′UTR-F	TAATACGACTCACTATAGGGCGCGGGTAACCCGGGATC
3′UTR-R	GGGCCGTTAGGAAATTACCTTAGTC

**Table 2 viruses-13-01523-t002:** Amino acid mutations in the continuous passages of two 3′UTR substituted chimeric viruses.

Virus	Passage	Amino Acid at the Indicated Position in Protein						
		E0	E2		NS4B		NS5B	
		476	745	979	2494	2671	3342	3431
vC		S	T	M	N	L	P	G
vC/SM3′UTR	p3							
	p11			K/M				
	p21			K		M	T	
	p31	R		K		M	T	S
vC/b3′UTR	p3							
	p11						P/T	
	p21			M/K			T	
	p31	R	I	K	H		T	
vShimen		R	T	R	N	L	T	G

**Table 3 viruses-13-01523-t003:** Fever response and viral replication in the spleens of the rabbits inoculated with viruses.

Inoculum	Dose (TCID_50_)	Fever Response				Viral Replication	
		No. with Fever/Total	No. of Hours to Onset	Duration (Hours)	Maximum Average Temperature (°C)	No. with Viral Replication/Total	Viral RNA Copies in the Spleens (Copies/μg RNA) *
vC/SM3′UTR	10^4^	3/3	30	10	41.0	3/3	(1.76 ± 0.42) × 10^4^
vC/SM3′UTR_P162T_	10^4^	3/3	30	8	40.3	3/3	(3.48 ± 0.97) × 10^4^
Negative control	/	0/2	/	/	/	0/2	/

* Data are presented as mean ± SD of 3 individual rabbits. Statistical analysis was performed using Student’s *t*-test. *p* < 0.05.

## Data Availability

The data presented in this study are available on request from the corresponding author.

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
