# Peer review of "Proline to Threonine Mutation at Position 162 of NS5B of Classical Swine Fever Virus Vaccine C Strain Promoted Genome Replication and Infectious Virus Production by Facilitating Initiation of RNA Synthesis"

_viruses, 2021, doi:10.3390/v13081523_

Round 1

Reviewer 1 Report

This original article describes a very elegant and successful approach aimed at discovering commercially interesting mutants of Classical Swine Fever Virus (CSVF) that are characterized by enhanced replication, capsid assembly and an overall increase in average specific productivity for infectious virus particles. The newly developed CSVF mutants described in this article constitute a major improvement for upstream CSVF viral vaccine manufacturability.

Starting point for the discovery of these productivity-enhanced mutants was the previously published finding that the 3’UTRs of CSVF are considered to play a prominent role in translation and replication of viral RNA and that a 12nt insertion within the 3’UTR of CSVF C-strain had a major decreasing effect on virus titer compared to the titers achievable with CSVF wild type strains. In this study the authors created chimeric mutants of CSVF ORFs and different 3’UTRs from other related Flaviviridae known to reach high titers.

Most interestingly, the authors not only corroborated their hypothesis that the chimeric mutants would show a much higher productivity but also discovered that the rescued high productivity mutants shared a common adaptive Pro-to-Thr mutation (P162T) located in the index finger subdomain of NS5B.

I have just two very minor suggestions for improvement. It would be nice to include data for the wild type CSVF strain and the CSVF C-strain into figure 1 c just to enable the reader to get a better idea of the quantitative increase in virus titer achieved by the evolved chimeric mutants. Also it might be nice to include an additional figure into the discussion showing a schematic overview of the proposed functional regulation between residue 162 in the index finger and residue 392 in the middle fingertip of NS5B and their relation to RNA template and NTP binding.

Author Response

We appreciate the critical comments from you. We have carefully considered the comments and revised the manuscript accordingly. We provided point-by-point responses to your comments.

Reviewer 2 Report

The manuscript by Huining Pang and colleagues titled “Proline to threonine mutation at position 162 of NS5B of classical swine fever virus vaccine C strain promoted genome replication and infectious virus production by facilitating initiation of RNA synthesis” described the P162T mutation at NS5B of classical swine fever virus vaccine exhibits enhanced RdRp  activity and thus helps in viral genome replication and infectious virus production. While the manuscript represents an interesting finding and has a decent amount of RdRp activity in vitro evidences and data, there are major considerations that prevents the manuscript’s publication in its current form.

Major considerations:

  1. The major issue in this manuscript is the lack of enough in vivo evidences to support the in vitro The analysis in the study was mainly based on the RdRp in vitro assays of Pro-to-Thr mutation (P162T) at position 162 of NS5B of the generated mutant viruses. While this is a valid analysis but it requires further verification in the biological system with in vivo studies in lab animals/experimental infection in pigs to confirm/prove your findings, that viral genomic RNA synthesis by changing initiation efficiency.
  2. Fig 1c: the total RNA copy number was measured till 12 hours post infection(PI). Did you also look beyond 12 hr PI? If yes, the data could be included to give broader picture of the infectious virus.
  3. Fig 1c & 2b: What is the rationale of choosing only two time points (6h, 12h, PI) for rescued chimeric CSF viruses whereas three time points (6h, 12h and 24h, PI) for chimeric CSF viruses with NS5B p162 T mutation?
  4. Fig 1b: The author has NOT provided the proper negative controls in IF (un infected PK-15 cells), so that the uninfected and infected cells can be clearly differentiated.
  5. Fig 1c & 1d: The author could include the wild type virus control while checking the viral RNA copy numbers and the virus infectious particles titer of the rescued chimeric viruses.
  6. Fig 1e: At the same time one step group curve can be included to compare the growth kinetics of the wild type virus and the rescued viruses. This would add more details to the rescued virus particles.
  7. Fig 6b: The color coded icons can be included as legends. How many times the RdRp assays were repeated and obtained the similar results?

Minor consideration:

  1. What is the detection limit / threshold limit of qRT-PCR? Did you use any housekeeping gene during the assay?
  2. SDS-PAGE gel pictures for the purified NS5B and mutant viruses could be included in the supplements.
  3. Future directions for the present findings and the short comings of your research may be included in the discussion.
  4. Fig 1b: the author used anti-NS3 antibody for detection of viral protein expression by IF and immune histochemical staining. Have you tried IF by using mAb against different proteins? If yes, you can show the data as supplements. Though NS3 plays an important role in stimulating the humoral and cellular arms of immune system but it is not effective against lethal challenge.
  5. Fig 2C: The graphs for virus titers vC/SM3’UTRP162T and vC/b3’UTRP162T mutants, appears to be identical however there is a clear visible difference in Viral RNA copy numbers. Please check the values again and confirm.
  6. The discussion section may be rewritten to include more references on the CSFV.
  7. The manuscript needs to be revised for the type errors like

Page 14 line 447: ‘seems’  instead of seem.

Page 8 line 291: missing text, 21st passaged?

Author Response

(The authors gave the same response as above.)
